# Anorectal and Genital Mucosal Melanoma: Diagnostic Challenges, Current Knowledge and Therapeutic Opportunities of Rare Melanomas

**DOI:** 10.3390/biomedicines10010150

**Published:** 2022-01-11

**Authors:** Margaret Ottaviano, Emilio Francesco Giunta, Laura Marandino, Marianna Tortora, Laura Attademo, Davide Bosso, Cinzia Cardalesi, Antonietta Fabbrocini, Mario Rosanova, Antonia Silvestri, Liliana Montella, Pasquale Tammaro, Ester Marra, Claudia Trojaniello, Maria Grazia Vitale, Ester Simeone, Teresa Troiani, Bruno Daniele, Paolo Antonio Ascierto

**Affiliations:** 1Oncology Unit, Ospedale del Mare, 80147 Naples, Italy; laura.attademo@gmail.com (L.A.); davidebosso84@gmail.com (D.B.); cinzia.cardalesi@gmail.com (C.C.); antonietta.fabbrocini@gmail.com (A.F.); rosanovamario@hotmail.com (M.R.); antonia.silv@libero.it (A.S.); b.daniele@libero.it (B.D.); 2CRCTR Coordinating Rare Tumors Reference Center of Campania Region, 80131 Naples, Italy; marian.tortora@gmail.com; 3Oncology Unit, Department of Precision Medicine, Università degli Studi della Campania Luigi Vanvitelli, 80131 Naples, Italy; emiliofrancescogiunta@gmail.com (E.F.G.); teresa.troiani@unicampania.it (T.T.); 4Department of Medical Oncology, IRCCS San Raffaele Hospital, 20132 Milan, Italy; marandino.laura@hsr.it; 5ASL NA2 NORD, Oncology Operative Unit, “Santa Maria delle Grazie” Hospital, 80078 Naples, Italy; liliana.montella@aslnapoli2nord.it; 6General Surgery Unit, Ospedale del Mare, 80147 Naples, Italy; pasqualetammaro81@gmail.com (P.T.); estermarra9@gmail.com (E.M.); 7Unit of Melanoma, Cancer Immunotherapy and Development Therapeutics, Istituto Nazionale Tumori IRCCS Fondazione Pascale, 80131 Naples, Italy; claudia.trojaniello@gmail.com (C.T.); dott.mariagrazia.vitale@gmail.com (M.G.V.); ester.simeone@gmail.com (E.S.); paolo.ascierto@gmail.com (P.A.A.)

**Keywords:** mucosal melanoma, rare tumors, rare melanoma, vulvar melanoma, vaginal melanoma, penile melanoma, anorectal melanoma, KIT mutation, BRAF mutation, immune checkpoints inhibitor, targeted therapy, referral center, multidisciplinary approach

## Abstract

Mucosal melanomas (MM) are rare tumors, being less than 2% of all diagnosed melanomas, comprising a variegated group of malignancies arising from melanocytes in virtually all mucosal epithelia, even if more frequently found in oral and sino-nasal cavities, ano-rectum and female genitalia (vulva and vagina). To date, there is no consensus about the optimal management strategy of MM. Furthermore, the clinical rationale of molecular tumor characterization regarding BRAF, KIT or NRAS, as well as the therapeutic value of immunotherapy, chemotherapy and targeted therapy, has not yet been deeply explored and clearly established in MM. In this overview, focused on anorectal and genital MM as models of rare melanomas deserving of a multidisciplinary approach, we highlight the need of referring these patients to centers with experts in melanoma, anorectal and uro-genital cancers treatments. Taking into account the rarity, the poor outcomes and the lack of effective treatment options for MM, tailored research needs to be promptly promoted.

## 1. Introduction

Mucosal melanoma (MM) is a rare type of tumor, being less than 2% of all diagnosed melanomas [1], comprising a variegated group of cancers arising from melanocytes in virtually all mucosal epithelia, even if more frequently found in oral and sino-nasal cavities, ano-rectum and female genitalia (vulva and vagina) [2]. Differently from cutaneous melanoma (CM), the aetiology, risk factors and pathogenesis of MM are poorly understood, thus explaining the lack of effective treatment options, the undesirable response rates and the extremely poor prognosis. Indeed, due to the rarity and the consequent paucity of literature data about efficient therapeutic strategies, as well as the deferred diagnosis often at an advanced stage, this melanoma subtype is characterized by a worse prognosis than melanoma arising from the skin [3]. Here, we summarize the current state of knowledge regarding anorectal and genital MM (penis, vulva, vagina) as models of rare melanomas deserving of a multidisciplinary approach, with the involvement of experts in the field of CM, urogenital and anorectal malignancies. In doing so, we focus on current gaps in our comprehension of anorectal and genital MM pathogenesis and treatments, passing through epidemiological features, risk factors, molecular characterization and local and metastatic disease management and promoting both the inclusion of patients in clinical trials and the development of tailored research.

## 2. Data Sources

Data on anorectal and genital mucosal melanoma have been collected through the search tools PubMed and Web of Science (date of last search 8 November 2021). A combination of Medial Subject Headings (MeSH), Major terms (MeSH heading that is of major importance in an article) and free text words was defined. We used the search including the following terms: mucosal melanoma, anorectal melanoma, ano-rectum melanoma, vulvar melanoma, genital melanomas, vulvovaginal melanomas, penile melanoma, penis melanoma, glans melanoma, urethral melanoma, BRAF, KIT and NRAS. Additionally, we incorporated several manuscripts using reference lists of articles found via electronical search. The Italian, European, American and British oncological guidelines of both anorectal, penis, vulvar cancer and cutaneous melanomas have been consulted.

## 3. Epidemiology and Risk Factors

Anorectal melanoma is rather an unusual tumor, accounting for less than 5% of all anal cancers and approximately 1% of all melanomas, thus representing one of the most frequent kinds of MM (25–30%) [4]; the main localization is the anal canal, while the anal verge is less frequently affected [5]. Concerning the median age of onset, anorectal melanomas are diagnosed in older patients compared to CMs (about 75 vs. 50 years), and, differently from cutaneous ones, they are more common in female than male [6].

Vulvar and vaginal melanomas are less frequent than anorectal ones, since their incidence is below 0.5 cases per 1 million women per year, accounting for less than 1% of all melanomas in the female sex (globally 15–20% of mucosal melanomas) [1,7]. Vulvar melanomas are more common than vaginal ones, with labia majora as the principal involved site and mainly diagnosed in postmenopausal Caucasian women [8]. Penile melanoma is an extremely rare entity, being less than 1% of all penile cancers and less than 0.1% of all melanomas in the male sex, mainly affecting older men (60–70 years) [9]. The risk factors are certainly different between mucosal and CMs since the latter are mainly dependent on UV-induced mutations [10]. The absence of UV mutational signature in MMs, since they arise in protected sites, is also responsible for their lower frequency in BRAF mutations [11], as explained below. Several carcinogens have been studied for assessing their role in MM onset, such as product of tobacco combustion and human viruses (e.g., human papilloma virus—HPV—and human herpes virus—HHV); however, no clear correlations have been pointed out to date [2]. Concerning anorectal melanoma, HIV infection could increase the risk of its development [12], whilst inflammatory diseases, such as Lichen Sclerosus, have been observed as a potential risk factor for vulvar melanoma [13]. Genetic predisposition could probably play a role in some cases of vulvar and vaginal melanoma, but a definite association with hereditary syndromes or diseases currently lacks for either anorectal or genital melanomas [2].

## 4. Biology and Genetic Alterations

Melanocytes are not exclusively located on the skin; they originate from multipotent neural crest cells in the neural ectoderm, then migrate to reach epidermis but also mucosae (especially those near the skin), eyes (uvea) and other inner organs such as leptomeninges [14]. It is common sense that MM arises from melanocytes localized in mucosae, but its pathogenesis is currently unknown; some authors have also suggested a putative origin different from melanocytes, but evidence is scarce so far [4]. Indeed, MM is widely recognized as a different entity than a cutaneous one, also from the molecular point of view.

First of all, the analysis of MM specimens showed lower single-nucleotide variant (SNV) and insertion/deletion (indel) mutation burden, which seems to be even lower in anorectal and genital melanoma [15] compared to CM [16]. The most important detected mutational signature is the age-related one, relying on the spontaneous deamination of 5-methylcytosine, consistent with the low mutation burden and the older age at diagnosis [15]. Moreover, rearrangement signature could identify a subgroup of MM characterized by amplification in *TERT*, *MDM2*, *CCND1* and *CDK4* genes, likely as a result of positive selection during tumorigenesis [15]. Concerning specific genetic alterations, data from the GENIE project of the American Association for Cancer Research [17,18] revealed that the five most commonly altered genes in MM are *NF1*, *TP53*, *KIT*, *CDKN2A* and *NRAS*. However, the most frequently reported alterations are *CDKN2A* losses, *NRAS* mutations, *KIT* mutations, *TP53* mutations and *TP53* missense alterations. *BRAF* activating mutations are definitely lower in MMs, than in cutaneous one, without differences according to anatomical site. In particular: BRAF codon V600 mutations are less than 10%, while non-canonical BRAF mutations, namely K601, L597, D594 and G469 are much more frequent [19,20]; BRAF fusions, described in around 5% of triple wild type (*BRAF*, *NRAS*, *NF1*) CMs, are not related to UV exposure [21] and have been found in a similar percentage in mucosal ones [19]. *KIT* gene, encoding for the transmembrane tyrosine kinase receptor KIT, has been found to be mutated in less than 20% of MM, most mutations localized at exon 11 and 13 [18,22,23]. The location of the primary tumour seems to be not indifferent, since its alterations are more common in anorectal and genital areas, according to Beadling et al. [24]; among these sites, however, an imbalance was observed, with a higher prevalence of KIT mutations in vulvar and penile melanoma and an absence in vaginal melanoma, suggesting different biological processes [25]. *NRAS* activating mutations, commonly affecting exon 2 (codon G12 and G13) and exon 3 (codon Q61), are less common in MMs than cutaneous ones, being less than a half (about 10%) [19]. Interestingly, fewer than a half of mutations are localized in codon 12 and 13, differently from CM, in which these mutations are significantly lower, probably reflecting differences in biological behaviour, since codon 12 and 13 activating mutations are weaker than codon 61 ones [20]. *NF1* is the most commonly altered gene in MM (about 20%), mainly through non-sense or frameshift mutations, that lead to a loss of function. The inactivation of the NF1 protein results in a deregulated and hyper activated MAPK pathway [26]. *CDKN2A* gene encodes for two proteins, p16 (also known as p16INK4a) and p14arf, which act as tumor suppressors and are disabled in many types of human cancer [27]. Loss of CDKN2A is the most frequently observed (epi)genetic alteration in MM (observed in more than half of patients) [28]; however, alterations in cell cycle regulator genes other than CDKN2A have been described in MM: *CDK4*, which has also been shown to be altered in a high percentage of acral melanomas [29], and *CCND1* genes were found as mutated (mostly amplified) in a not so small number of cases [15,18]. *MDM2*, encoding for the homonym ligase responsible for TP53 ubiquitination, and *MDM4* genes amplifications have been correlated with a poor response to immunotherapy [30].

Less frequent genetic alterations affect genes involved in genomic preservation (*TERT*) and signaling pathways (*ALK*, *NTRK* and *SPRED1*). *TERT* is a gene encoding for the catalytic subunit of telomerase, which maintains telomeric integrity, ensuring cell immortality in human cancer [31]. Differently from CM, in which mutations occur mainly in the promoter region [32], the TERT alterations described in MM do not affect the promoter but consist in copy number variations, ultimately determining overexpression and stopping senescence [15,33,34]. *ALK* fusions, which are well-known tumors driven in lung adenocarcinomas, have been identified in ~11% of CMs, and, interestingly, MMs could also harbour such alterations [19,35]. NTRK includes a family of three genes encoding tropomyosin receptor kinases (TRK): TRKA, TRKB and TRKC receptors and corresponding to the NTRK1, NTRK2 and NTRK3 genes, respectively. These receptors are involved in the development and function of neuronal tissue [36]. Gene fusions are the major molecular aberration involving NTRK in tumorigenesis and have been detected in several histologically different cancers, including many different epithelial tumors, glioblastomas and sarcomas. In melanoma, NTRK fusions seem to be relatively common in spitzoid melanoma, with a prevalence of 21% and 28.5% [37,38]; however, the prevalence in cutaneous or MM is <1% [39,40]. Lezcano and colleagues investigated the prevalence of NTRK fusions in a group of 751 patients with metastatic melanoma, discovering NTRK fusions in three cutaneous and one MM (anal MM), indicating a frequency of 0.8–0.9%. Of note, all melanomas with NTRK fusions were thicker than 2 mm and remarkable for their epithelioid cellular pattern [40]. The *SPRED1* gene encodes for a tumor suppressor protein, which inhibits the MAPK pathway by interacting with NF1; *SPRED1* loss acts as a molecular driver in MM [41] and has been identified in less than 8% of MM patients, possibly representing a distinct biological entity [15]. Interestingly, mutations in the *SF3B1* (Splicing factor 3B subunit 1) gene are more frequent in anorectal and vulvo-vaginal melanoma than in other mucosal sites [42,43], but the potential implications of this discovery are unknown. (Figure 1).

## 5. Diagnosis, Prognosis and Loco-Regional Treatments

### 5.1. Anorectal Mucosal Melanoma

Anorectal mucosal melanoma (AMM) is a rare mucosal disease with a particularly aggressive behaviour compared to cutaneous melanoma at an equal stage. Despite its rarity, AMM is the second most common subtype of mucosal melanoma after ocular melanoma, representing approximately 0.6–1.6% of all melanomas and approximately 1–4% of all anorectal malignancies [44]. The majority of patients are Caucasian, with the highest incidence during the sixth and seventh decades. There is a slight female preponderance [45]. Unfortunately, when diagnosed, one third of the patients present metastatic disease not amenable of surgical cure; thus, the median overall survival after diagnosis is between 8 and 19 months [5], and the 5-year survival is approximately 20% for loco-regional disease, decreasing at 0% in the case of advanced disease [46]. A poor overall survival in AMM has been associated with male gender, perineural invasion, depth infiltration of the rectal wall, lymph node metastasis and distant metastasis [47,48]. It has been hypothesized that AMM usually originates from melanocytes present in the transitional zone of the surgical anal canal and tends to spread along the submucosal plane draining to the inguinal and inferior mesenteric lymph nodes [47,48]. Indeed, AMM is staged according to the disease dissemination. Local disease is defined as stage I, regional lymph node disease as stage II and metastatic disease as stage III. It is still controversial if the tumor size and tumor thickness are prognostic factors for AMM. Thibault et al. reported that patients with a tumor thickness ≤2 mm have better survival than patients with lesions >2 mm [49], while Goldman et al. described a correlation between overall survival and tumor size, with greater overall survival for patients with tumors ≤2 cm [50]. Nevertheless, more recent literature data reported tumor size and thickness as not associated with disease-specific survival [51,52]. The most relevant prognostic factor in AMM remains the disease stage, and, consequently, the early diagnosis should be the key to improving survival outcomes in these patients [53,54]. The bad prognosis depends mainly on the absence of specific clinical symptoms that can delay the initial diagnosis; thus, almost 60% of patients have already disseminated the disease at initial presentation. Unfortunately, to date, due to the rarity and the aggressiveness of this disease, with such a limited number of affected patients, a standard diagnostic and therapeutic approach has not yet been well established. Experts’ reports suggest referring to a colorectal surgeon as soon as possible (e.g., 2 weeks) for patients with any of the following symptoms or signs: rectal bleeding, palpable lymph nodes associated with anal symptoms, atypical haemorrhoids, polyps especially if associated with pruritus, change in bowel habit and tenesmus. Patients with anorectal irregularly outlined pigmented or non-pigmented macule, papule, patch or nodule with or without ulceration should be referred to a dermatologist with experience in pigmented lesions [54]. Since the presenting symptoms of AMM may be similar to those of rectal cancer, surgeons should carefully inspect the anal margin, as not all melanomas are pigmented, and proceed for the histopathological diagnosis with excision or punch biopsy, depending on the size and site of the lesion, for the final diagnosis. Patients who present with anorectal lesion and palpable groin lymph nodes should have pathological confirmation either by fine needle aspiration (FNA) or core biopsy of the suspicious nodes [54]. The loco-regional staging should be completed with proctoscopy and pelvis magnetic resonance imaging (MRI). Although surgery is the mainstay treatment for localised AMM, the most appropriate strategy is still controversial and debated between two approaches: the radical one with the abdominoperineal resection (APR) and the conservative one with the wide local excision (WLE). Despite the surgical approach’s choice, the treatment for AMM should be planned in centers regularly performing complex anorectal surgery and regularly managing complex melanoma within a multidisciplinary experts’ team [54]. If radical resection is being evaluated, FDG-PET plus a whole-body CT scan, together with a brain CT scan or MRI, should be performed to exclude disseminated disease, given the common pattern of the metastases of the disease (brain, lung, liver and bone) [55]. Radical resection with palliative intent, in the presence of distant metastases, may still be an option, but only after multidisciplinary assessment and taking into account the patient’s willingness and preserving the quality of life [56]. In the past, APR was the most performed surgical procedure to provide a better local control, but, as its impact on survival outcomes is still not clarified, the goal of the surgical approach should be obtaining negative margins while preserving sphincter function [4,48,51,56,57,58]. Tumor thickness is a strong predictor factor for the risk of local recurrence and should be considered to plan therapeutic procedures. Tumor thickness below 1 mm can be performed by local sphincter-saving excision with a 1 cm safety margin. Tumor thickness between 1–4 mm should be excised by a sphincter saving excision and a 2 cm safety margin, and tumors with thickness above 4 mm or with the involvement of sphincters should be treated with APR. In the event of R1 margins (margin < 1 mm), a repeat local excision or radical resection should be performed to obtain an R0 margin [48,54,57,58,59,60,61,62]. Unfortunately, despite the surgical loco-regional control, most patients progress to metastatic disease. Stoidis et al. assumed that systemic disease dissemination, via lymphatic and haematogenous, is an early event in tumorigenesis. Lymphatic spread to mesenteric nodes is more common than to inguinal ones, while lungs, liver and bones are the most frequent sites of distant metastases [57]. Prophylactic lymph node dissection (LND) has no adjunct value, and the therapeutic LND should be performed only in the presence of positive inguinal nodes. Sentinel lymph node biopsy (SLNB) is only recommended in the light of directs adjuvant treatment or clinical trial eligibility; after a positive sentinel node, there is the option of following the patient by clinical examination and ultrasound. However, the nodal dissection should be planned after multidisciplinary discussion [54]. Because survival is determined by distant disease dissemination, many centers with expertise in anorectal cancers surgery have adopted sphincter-sparing excision for primary tumor control. However, considering the higher rates of local failure (~50%), the University of Texas MD Anderson Cancer Center reported the outcomes of 54 patients with localized AMM treated with definitive local excision with or without SLNB or LND, followed by local or extended radiotherapy (RT) (25–36 grays in 5–6 fractions): in this series, combined sphincter-sparing local excision and RT was a well-tolerated approach that provided effective local control, while the inclusion of the inguinal lymph node in the RT fields did not improve outcomes and was associated with an increased risk of lymphedema [63]. Despite the experiences of very small patients’ series, the role of adjuvant therapy still needs to be clarified. Historically, the response of AMM to radio and/or chemotherapy has been very poor. No systemic therapy regimen or radiotherapy for adjuvant setting in AMM is to date considered as a standard of care. A proposed flowchart for the management of AMM is provided in Figure 2.

### 5.2. Vulvo-Vaginal Mucosal Melanoma

Gender plays an important role in the epidemiology of MM due to the fact that genital mucosal melanomas are less frequent in men than in women. Among female patients, the vulva is the predominant location of melanoma, the vagina being the least affected [63]: in detail, primary melanoma of the vagina (PMVa) are four to nine times less common than primary melanomas of the vulva (PMVu) [64,65]. It should be noted that location also has a prognostic significance, since the 5-year overall survival rate for PMVu is more than double respect to PMVa (about 45% and 20%, respectively) [64,65,66,67,68]. This prognostic difference depends on several factors: the delayed diagnosis for PMVa due to the relatively high rate of amelanotic tumors, the anatomical proximity to the vulvovaginal venous plexus and the different tumor biology suggesting a different ontogenetic development. Indeed, the molecular characteristics of PMVa are markedly different from those of PMVu, especially regarding the KIT mutations, as previously mentioned. Aulmann et al. reported the molecular characterization of 65 cases of vulvo-vaginal melanoma (VVMM), finding no BRAF mutations but NRAS mutations and KIT amplifications in 12% of both vulvar and vaginal tumors [69]. In agreement with these results, Rouzbahman et al. found BRAF mutations in 8%, KIT mutations in 28%, NRAS mutations in 28% and TP53 mutations in 8% of PMVu cases and only one case of PMVa with TP53 mutation [70]. Despite the molecular differences, major surgery, radiotherapy and systemic treatments, VVMM has a poor prognosis [71]; this is mainly due to the high metastatic potential of these tumors, explained by the rich vascular and lymphatic system of the female urogenital tract mucosa, thus leading to delayed diagnosis [72]. Concerning the staging system for the VVMM, the International Federation of Gynecology and Obstetrics (FIGO) staging system is not applicable, so clinicians should adopt the American Joint Committee on Cancer (AJCC) melanoma staging system, which is obtained by the combination of several clinico-pathological characteristics (tumor thickness, tumor ulceration, status of regional lymph nodes, site(s) of distant metastases and serum lactate dehydrogenase) and is to date the most important and reliable predictor of recurrence-free survival [73,74,75]. The onset of VVMM commonly occurs in the sixth and seventh decades of life, vulvar and vaginal lesions, pruritus, discomforts or bleeding being the initial symptoms [76,77]. The exact aetiology of VVMM is unknown [78]; however, it is obvious that, differently from cutaneous melanomas, ultraviolet radiation is not the causal factor in VVMM [79]. It has been speculated that these mucosal malignancies arise from melanocytes located aberrantly in vaginal epithelium [78,80], which can be found normally in the basal layer of vaginal epithelium in 3% of healthy women [81]. It has been supposed that active junctional changes are the initial development stage in malignant MM [81]. Although VVMM might arise anywhere, PMVa is primarily detected in the lower one third (34%) and mostly on the anterior wall (38%) of the vagina [73,82], presenting as single or multiple and pigmented or non-pigmented [83]. Additionally, most PMVas are polypoid and ulcerated [78,84]. Epithelioid is the most common histotype of PMva (55%), followed by mixed (28%) and spindle (17%) type [78,84,85]. Concerning PMVu, the most commonly affected sites are the labia majora (about 50%), the labia minora (less than 20%), clitoris (about 10–15%) and Bartholin’s glands (less than 5%) [86,87], whilst the main histological types are mucosal lentiginous, nodular and superficial spreading [88]. Disease relapse may occur locally—in the pelvis—or at a distance—in the lungs, liver, bones and, less frequently, the brain. It is common to observe the concomitant onset of both local recurrence and distant metastases [70]. Despite several treatment options for VVMM being currently available, a standardized and effective treatment algorithm has not been defined yet [6]. Experts’ reports suggest referring to a gynaecological oncology team and to a dermatologist with an interest in pigmented lesions as soon as possible (e.g., 2 weeks) for patients with any of the following symptoms or signs: pigmentation; persistent itching with pigmentation; bleeding lesion; irregularly pigmented or non-pigmented macule(s), papule(s), patch(es) or nodule(s) with or without ulceration; groin lymph node(s) enlargement associated with a vulvo-vaginal pigmented lesion; obstruction of urethral meatus with pigmented lesion(s) [54]. For small vulvar lesions highly suggestive of MM, an excisional biopsy should be performed, whereas for larger lesions, an incisional biopsy or punch biopsy should be done. Patients who present with vulvo-vaginal lesion(s) and palpable groin lymph nodes should have pathological confirmation either by FNA or by core biopsy of the suspicious nodes [54]. The loco-regional staging should be completed with speculum examination, cystoscopy in the case of urethral involvement and pelvis MRI. In the case of eligibility for radical resection, similarly to AMM, FDG-PET and WB CT scans, together with either a brain CT scan or brain MRI, should be performed to exclude disseminated disease, given the high risk of lung and brain metastases [54,89]. Surgery remains the primary treatment of choice in the case of localized disease. As for AMM, for VVMM, the surgery approach is still matter of debate. The spectrum of surgery ranges from conservative (WLE) to radical (vulvectomy/vaginectomy and pelvic exenteration). However, the role of radical surgery as a primary treatment for VVMM remains unjustified considering the serious and long-term morbidity, sexual dysfunction, psychological burden and the absence of a clear survival benefit [73,90,91]. For many years, radical vulvectomy defined as “the removal of the entire vulva until the deep fascia of the thigh, the periosteum of the pubis and the inferior fascia of the urogenital diaphragm” was the standard treatment for squamous-cellular vulvar carcinoma and was also assumed for PMvu [80,91]. WLE, defined as the excision of the neoplasm with wide tumour-free surgical margins, has been proposed as an alternative and, to date, is the selected primary surgical treatment, considering the similar survival rates compared to radical vulvectomy [8,72,90,91,92,93,94,95,96]. Nevertheless, knowledge about the optimal surgical margins of the WLE for VVMM is lacking. Irvin et al. proposed identical margins of cutaneous melanoma for VVMM (0.5 cm for in situ melanomas, 1 cm for lesions up to 2 mm thick and 2 cm margins for melanomas more than 2 mm thick) and neither found margins wider than 2 cm to improve survival [80]. In VVMM, elective LND is not recommended given the lack of survival benefit [97,98,99]. SLNB may add information on regional node involvement. Since in VVMM positive pelvic nodes are rarely detected, in cases of negative inguinal nodes, SLNB status is thought to predict the status of the further nodes. However, evidence on this matter is scarce, and a SLNB should be recommended only if it impacts on adjuvant treatment or clinical trials entry choice [54]. Radical resection with palliative intent in the presence of distant metastases may be still an option but only after clearly documented discussion among the multidisciplinary team and considering the patient’s willingness [54]. Adjuvant local RT for VVMM have shown to improve local control with no impact on overall survival [93,100,101,102]. However, in cases of tumour positive or narrow margins, adjuvant RT may be reasonable [103]. Neoadjuvant RT has been proposed in patients with head and neck melanoma not amenable to surgical resection [102,103,104,105], but no data are available for neoadjuvant RT in VVMM.

The activity of neoadjuvant anti-CTLA4 antibody ipilimumab with concomitant RT has been reported in four patients with VVMM, obtaining 1 stable disease, 2 partial responses and 1 complete remission. Nevertheless, the authors of the reported case series conclude that the combination of RT and immunotherapy as neoadjuvant treatment should only be deserved in a clinical trial setting [106]. RT of the groin after LND has not been investigated in VVMM; however, it is reasonable considering regional RT in the case of macroscopic, unresectable lymph node involvement [107]. Neoadjuvant use of (bio)chemotherapy aiming at the reduction of tumour mass has been reported in one PMVu and two PMVas. In the first case, carboplatin and paclitaxel in combination with the anti-angiogenetic drug bevacizumab achieved considerable reduction of the 5 cm large melanoma, making resection possible and improving the skin graft [108]. Adjuvant chemotherapy did not show any survival benefit in VVMM [100,108,109]. A proposed flowchart for the management of VVMM is provided in Figure 2.

### 5.3. Penile Mucosal Melanoma

Primary malignant melanomas of the glans, penis and urethra, more accurately described as penile mucosal melanoma (PMM), are rare neoplasms occurring in <0.1% of all melanomas [9]. The literature regarding penile mucosal melanoma (PMM) is limited and based on case series and reports including a small number of patients and often lacking a long follow up and key information. A review considering the five largest series of penile and urethral melanomas published between 1962 and 2012 [110,111,112,113,114] reported a median overall survival of 28 months (range 2–276) and a 5-year survival of approximately 10%, with better outcomes for patients with stage I and a median invasion depth of 2 mm (range 0.3–2 mm) [9]. The diagnostic delay and the high incidence at diagnosis of vertical growth phase in glans, similar to nodular cutaneous melanoma, are some of the reasons that can explain the poor outcomes of this disease [9,111,115,116,117]. Depth of invasion is suggested as one of the most important risk factors for metastases, and a study including 66 patients with PMM suggested that the prognosis is not worse than that for cutaneous melanoma with comparable tumor thickness [110]. Typical symptoms of penile mucosal melanoma (PMM) are bleeding from a penile lesion, urethral discharge and haematuria, but fewer specific symptoms, such as lower urinary tract symptoms, dyspareunia and irritation, may also be present [54]. Nevertheless, patients are often asymptomatic at an early stage [117,118], which is one of the reasons for the diagnostic delay. PMM should enter in the differential diagnosis when irregularly pigmented or non-pigmented lesions (i.e., macule, papule, patch or nodule) with or without ulceration are detected on the penis or foreskin or when a palpable urethral lump or palpable inguinal lymphadenopathy are found during the clinical examination [54,117,119]. Glans melanoma is typically a coloured lesion, with ulceration in approximately 39% of cases [54]; it is located in the majority of cases (63%) on the dorsal or ventral part of the glans, while the location at the meatus and in the corona is less frequent, representing 28% and 9% of cases, respectively [9]. The patient’s referral to a urologist/penile cancer specialist or dermatologist with expertise in pigmented lesions is of utmost importance for a timely diagnosis of PMM and should be considered whenever the above-mentioned symptoms or signs are detected [54]. Both the urology and the melanoma multidisciplinary tumour board should be involved in the patient’s management. Unfortunately, as previously anticipated, diagnostic delay is common, and patients are often referred to other different specialists before diagnosis of the disease [54]. External clinical examination and palpation of inguinal lymph nodes represent the first steps for the local staging. A penile MRI with a pharmacologically induced erection could help in the surgical planning in order to determine the local extent [54,119]. Local staging should also be completed with a cysto-urethroscopy in cases of urethral involvement or proximity to the peri-meatal area [54]. CT of the thorax, abdomen and pelvis including the groins is used to exclude distant metastases, and MRI or CT of the brain should be also considered to exclude brain metastases. Total body FDG-PET and MRI of the brain are advised before planning a surgery with curative intent [54]. Unfortunately, due to the rarity of the disease, there are no standardized guidelines for the staging of PMM [9]. Most authors use a three-stage system for PMM as for AMM, where stage I includes a disease confined to the penis, stage II includes melanoma with lymph node involvement and stage III a disseminated metastatic disease [115,120]. The UK national guidelines on ano-uro-genital mucosal melanoma and other authors suggest the use of the AJCC TNM Staging System; of note, the 8th edition of the AJCC staging system does not include a site-specific staging. However, the presence of nodal/distant metastases should be recorded using the N and M of the AJCC/TNM system as though the melanoma were a carcinoma [54]. However, it remains unclear whether this accurately reflects the behaviour of PMM.

Depending on the size and the site of the lesion, excision biopsy or punch biopsy should be used for the diagnosis. In typical and small lesions, an excision biopsy is advised, while in larger lesions, a punch biopsy is an alternative [119]. In cases of palpable lymph nodes, ultrasound and FNA or core biopsy of the nodes are recommended [54,119]. The surgical treatment of primary PMM depends on the stage and the location of the lesion [119]. When radical resection is considered, surgery should be performed in referral penile cancer centers and after a comprehensive evaluation of the patient, including assessments of comorbidities, the patient’s quality of life and urinary and erectile function [54]. As for cutaneous melanoma, the mainstay of mucosal melanoma treatment is a wide excision with negative margins [9]: a R0 margin (microscopically clear >5 mm) should be the aim of the surgical procedure; however, deep margins >1 mm are accepted if a glansectomy is performed [54]. Due to the rareness of this disease, only small case series, often not controlling for confounding factors, comparing radical excision with local excision exist [111,112,113,114,115,116,117,118,119]. However, to date, radical surgery (partial or radical penectomy) does not seem to have an impact on overall survival. SNLB may be performed, as in the management of penis squamous carcinoma, and lymphadenectomy is indicated only in cases in which there is evidence of metastatic regional nodal disease [54]. In cases of R1 margins, local re-excision or radical resection is recommended in order to obtain R0 margins [54]; however, when a surgical procedure is not feasible, adjuvant RT may be considered to reduce the risk of local recurrence, even if there are no studies regarding adjuvant radiotherapy in this specific subtype [54,119]. As for the mucosal melanoma of other sites, primary mucosal melanoma is at high risk of local and distant relapse, and a close follow-up is recommended, especially in the first 3 years [119]. Unfortunately, the are no studies of adjuvant systemic therapy including more than one patient with penile mucosal melanoma [119,121]. A proposed flowchart for the management of PMM is provided in Figure 2.

## 6. Systemic Treatments for Anorectal and Genital Mucosal Melanoma

Worthy of mention is the only randomized phase II trial, including 189 Asian patients with completely resected stage II/III MMs, that compared temozolomide plus cisplatin chemotherapy and high dose interferon alpha-2b (IFN-A) therapy (HDI) with surgery alone. Lian and colleagues concluded that both adjuvant regimens were safe and well tolerated, with the temozolomide plus cisplatin regimen more likely to improve relapse free survival (RFS) and overall survival (OS) than HDI [median RFS: 20.8 vs. 9.4 months (*p* < 0.001); median OS: 48.7 vs. 40.4 (*p* < 0.01), respectively]. However, the authors pointed out that as a single-center trial with only Asian patients, the adjuvant therapy using chemotherapy or HDI in patients with MMs would have required more randomly controlled trials, never conducted, including Caucasian patients [122]. In the past decades, advanced MMs have also been commonly treated with systemic chemotherapies [123], such as cisplatin, vinblastine, and dacarbazine, or by immune agents such as IFN-A and interleukin 2 (IL-2); however, these treatments showed low effectiveness on response and survival. For inoperable/metastatic AMM disease, Kim et al. proposed a combination of temozolomide, cisplatin and liposomal doxorubicin, achieving promising results at the expense of the toxicity profile [124,125]. In advanced VVMM, the only study exploring (bio) chemotherapy is a case series of 11 PMVu and PMVa patients treated with a combination of cisplatin, vinblastine, dacarbazine, temozolomide, tamoxifen, IL-2 and IFN-A. In all cases, the median survival was 10 months, and 36% achieved a partial response, with a non-negligible percentage of adverse events [126]. More fragmented are the data reported about the use of (bio) chemotherapy in PMM, where cytotoxic agents have been employed mostly in adjuvant settings, with very unsatisfactory results [127,128].

Since conventional chemotherapies failed to improve survival outcomes in patients with advanced MMs, thus highlighting that the choice of systemic treatment should be guided by the most available contemporary data, the last part of this overview will be focused on current knowledge of immune check-points inhibitors (ICIs) and targeted therapies’ (TT) applications in MMs.

### 6.1. Immune Check-Points Inhibitors

Several efforts have been made over the last years to identify therapies that could directly act on patients’ immune systems in a way that restores or induces a response to cancer, thus leading to regulatory approvals of an entire new class of anti-cancer drugs, known as ICIs, for the treatment of a variety of malignancies. The first to be approved in 2011 was the anti-CTLA-4 antibody ipilimumab for the treatment of unresectable or metastatic cutaneous melanoma. Subsequently, the anti-PD-1 and anti-PD-L1 antibodies also received approval for several haematological and solid cancers [129]. With regard to MMs, specific randomized clinical trials of both anti-CTLA-4 and anti-PD-1/PD-L-1 antibodies are currently unavailable, but these drugs seem to be effective, according to some literature data and a systematic review [130], being part of the therapeutic opportunities for these rare malignancies. In a large French retrospective cohort of 229 MM patients treated with either ICIs or CT as first-lime treatment, 131 patients with AMM and VVMM were included; this study confirmed the superiority of ICIs on CT [131]. It is worth mentioning that in a pooled analysis of five clinical trials, the efficacy of nivolumab plus ipilimumab was greater than nivolumab alone, despite the activity being lower in MM than in CM [132]. Other trials also showed activity of nivolumab or pembrolizumab treatment in patients with mucosal melanoma, but without long-term follow-up [133,134,135]. Among these, an American multi-institutional, retrospective cohort analysis identified 25 adults with advanced acral melanoma and 35 adults with MM who received treatment with nivolumab or pembrolizumab as standard clinical practice through expanded access programs or published prospective trials, reporting in those with MM an objective response rate (ORR) of 23% (95% confidence interval CI, 10%–40%) and a median progression free survival (PFS) of 3.9 months after a follow up of 10.6 months [133]. In 85 patients with MM included in the post-hoc analysis of KEYNOTE-001, 002, 006 and treated with pembrolizumab 2 mg/kg every 3 weeks or 10 mg/kg every 2 to 3 weeks, the ORR was 19% (95% CI 11–29%), with a median duration of response (DOR) of 27.6 months (range 1.1 + to 27.6), a median PFS of 2.8 months (95% CI 2.7–2.8) and a median overall survival (OS) of 11.3 months (7.7–16.6) [134]. The combination of RT and pembrolizumab has been evaluated in a Korean retrospective study, where 23 patients between July 2008 and February 2017 received RT for primary or metastatic tumor mass with a median dose of 4 Gy per fraction (range 1.8–12 Gy); eleven patients were treated with RT alone, whereas 12 patients underwent pembrolizumab combined with RT and 8 patients with metastatic MMs treated with ICI alone during the same study period were included as a comparison group. After a median follow-up period of 17.4 months, the target lesion control rate at 1-year was significantly higher in the ICI+RT group than in the RT-alone group or ICI-alone group (94.1% vs. 57.1% vs. 25%; *p* < 0.05); no abscopal effect was observed, and treatment-related adverse events were not significantly increased in the combined treatment group compared with the RT-alone group (*p* > 0.05) [135]. A Dutch population-based study retrospectively evaluated 3080 patients—2960 affected by CM and 120 affected by MM—treated with ICIs between 2013 and 2017; among MM, 29 (24%) had VVMM. Interestingly, median OS was similar in the MM and VVMM groups (8.9 and 8.6 months, respectively), and no improvement was recorded for patients with MM diagnosed in 2013–2014 or 2015–2017, differently from what has been observed in CM patients, in which the OS increased from 11.3 months to 16.9 months, respectively [136]. Concerning the type of ICIs, an Italian and a French study—with 7 and 15 patients respectively—have reported better outcomes for VVMM patients treated with anti-PD-1 agents than those treated with anti-CTLA-4 [137,138]. The ICIs combination’s effectiveness and safety has been recently evaluated in the post hoc analysis of the CheckMate 067 trial, presented at the American Society of Clinical Oncology (ASCO) Congress 2020, in which 79 patients with previously untreated stage III or stage IV MM were included [139]. First of all, this 5-year analysis showed that the MM population differed from the intent to treat (ITT) population in having a higher proportion of women and a generally poorer long-term efficacy; however, similar safety outcomes, considering both the incidence of all grade and grade 3/4 of adverse events and a similar proportion of patients who received any subsequent treatment for both nivolumab plus ipilimumab and ipilimumab subgroups, were observed. Moreover, MM patients treated with the ICIs combination of nivolumab plus ipilimumab appeared to have more favourable survival outcomes than those treated with nivolumab or ipilimumab alone, with, respectively, a 5-year PFS of 29%, 14% and 0% and a 5-year OS of 36%, 17% and 7%. The addition of ipilimumab to nivolumab treatment seemed to increase the rate of complete responses and the duration of the response compared to nivolumab alone, reaching the greatest decrease in tumor burden with a median change of –45%. Noteworthy is that 9 of 28 patients in the combination ICIs group (32% of the total group) discontinued treatment and remained treatment-free while maintaining a response at 60 months from randomization, including three patients who were treated for ≤16 weeks; as well, in the nivolumab group, two patients remained on treatment with a response and one patient had an ongoing response after discontinuing treatment at 48 weeks and remained treatment-free. The most common reason for discontinuing treatment was toxicity for the combination group and progression for the single ICI group [139]. Another point of discussion is the lack of validated biomarkers for ICIs use in MM: in this regard, PET-FDG may have a role, but further studies are required [140]. Taking into account these data, the choice of systemic treatment should be always guided by the most updated evidence, the potential contraindication and the individual patient’s features. However, the combination, for instance, of anti-CTLA-4 and anti-PD-1/PD-L1 should be considered as the first option, after having carefully discussed the potential toxicities with patients.

### 6.2. Targeted Therapy

Considering the lack of robust data about the effectiveness and the safety of systemic therapies for advanced and metastatic MMs, clinicians should always keep in mind to search for any actionable driver mutations in these rare cancers, in a way to identify any TT potentially administered on a patient-by-patient basis. Currently, there are only few available TTs for MM clinical trials: BRAF, MEK, CDK4/6 and C-KIT inhibitors, with limited use in daily clinical practice. Moreover, as mentioned above, MM, differently from the cutaneous one, is deficient in dominant MAPK activating mutations [141]. Nevertheless, some efforts on investigating target strategies based on mutated genes in MM have been done, and, here, we report the most interesting relative literature data.

For the minority of metastatic MM patients who harbour BRAF mutations, combined inhibition of BRAF and MEK may represent a valid therapeutic opportunity, as this combination therapy already showed an impressive response rate and survival benefit in BRAFV600E/K positive cutaneous melanoma (CM) patients [142]. Indeed, a Japanese retrospective study evaluating BRAF and MEK inhibition also in MM with BRAF mutations reported similar response rates of those achieved for CM (64.3% vs. 76.5%) patients [143]. Furthermore, for patients with NRAS, NF1 or SPRED1 alterations, a potential therapeutic strategy may be represented by targeting the downstream protein MEK. An ongoing phase II trial investigating the safety and efficacy of the MEK inhibitor binimetinib reported a response rate in 20% of patients with NRAS mutation, including MMs [144]. Unfortunately, the acquired resistance through the reactivation of the MAPK pathway may also occur for MM patients treated with both the MEK inhibitor or the MEK inhibitor plus the BRAF inhibitor [21,145]. To overcome this resistance, it could be useful to inhibit downstream proteins like ERK or develop new molecules targeting aberrant MAPK signaling [3,146]. Moreover, MEK inhibitors have also been studied in combination with mTOR1/2, AKT or CDK4/6 inhibitors in preclinical models or in clinical trials of MM [147,148]. As previously reported, MM patients present more activating mutations or amplifications in the receptor tyrosine kinase c-KIT, providing a rationale for targeting C-KIT. Imatinib is the most widely investigated c-KIT inhibitor [149], followed by other approved C-KIT inhibitors such as sunitinib, dasatinib, nilotinib and masitinib, currently under investigation for c-KIT mutated melanoma in several clinical trials [150,151,152,153]. However, responses to c-KIT inhibitors depend on the type of mutations, as they are longer and higher for the mutations in exon 11 (L576P) and exon 13 (K642E) of the *KIT* gene [154,155], while *KIT* amplification is a non-responding genetic alteration (54% vs. 0% partial response, respectively) [153,156]. In support of this observation, FDG-PET responses were also correlated with the type of *KIT* mutations during c-KIT inhibitors treatment [157]. In a recent trial including 78 *KIT* mutated melanoma patients, the median overall survival for imatinib was 13.1 months, with an objective response rate of 21.8% [158]. As for BRAF and MEK inhibitors therapy, the issue of acquired resistance, unfortunately, also occurs for c-KIT blocking. Indeed, patients who underwent c-KIT inhibitors therapy after a (usually short) period of treatment response developed drug resistance, which leads to progressive disease [2,159]. The acquired resistance to c-KIT inhibitors is possibly due to pre-existing concomitant mutations in other oncogenes such as NRAS or secondary *KIT* mutations during the TT delivering. For example, melanoma cells, harbouring secondary A829P *KIT* mutations, are resistant to imatinib but still sensitive to nilotinib and dasatinib, while the T670I *KIT* mutation confers resistance to imatinib, nilotinib and dasatinib, preserving sensitiveness to sunitinib [160]. Considering the encouraging results of C-KIT inhibitors in MMs, more efforts should be made to investigate the mechanisms of acquired resistance, thus developing new blocking agents to overcome resistance and proposing chances of a cure for advanced MM patients with very few treatment options. Lastly, targeting NTRK fusions may represent a new chance for MM treatment, since selective inhibitors (entrectinib and larotrectinib) have demonstrated a disease response rate of more than 50% in tumors with NTRK fusions, regardless of histology [161,162]. Moreover, further studies focused on adjuvant targeted therapy based on gene sequencing results should be promoted [163].

## 7. Follow-Up

As for any type of cancer, for anorectal and genital MM, the primary aim of follow-up is to detect as soon as possible loco-regional or distant recurrences in an early stage to prolong survival [164]. Up until now, there have been no guidelines on AMM, VVMM and PMM follow-up, and schedules have been based on the clinical experience and practice rather than on evidence. The consultation of the respective national guidelines for CM is also recommended. For localized AMM, VVMM and PMM, it is reasonable to apply the follow-up scheme that comprises appointments every 3 months, during the first 3 years and every 6–12 months thereafter. Intervals between clinical visits and imaging exams may be tailored according to the individual risks and personal needs of the patient. Since loco-regional and distant recurrences, as well as late recurrences (>5 years), are frequent, a long-term follow-up plan should be scheduled [165,166]. The first post-operative appointment is needed to detect any occurrence of surgical complications and to evaluate eligibility for any adjuvant therapy or clinical trials. Adding imaging and a laboratory test should be required when suspicion is raised for a recurrence or distant dissemination. In cases of loco-regional or metastatic disease, the follow-up should include a CT scan of the thorax, abdomen and pelvis including the groin and a MRI or CT scan of the brain every 3 months for patients treated with ICIs and every 2 months for those treated with TT. It should be acceptable to extend imaging evaluation to 6 months up to year 5, and then annually up to year 10, in patients who have achieved a prolonged control disease [54]. A special consideration should be finally given for any need of psychological support, since anorectal and genital MM diagnosis and treatments lead to a considerable worsening of quality of life due to emotional, physical and social functioning, sexuality and body image impairment [167,168].

## 8. Conclusions

Anorectal and genital MMs are highly malignant entities that are very hard to diagnose, treat and study and even more complex and heterogeneous than CM from the molecular point of view. Rarity, data paucity and poor prognosis may have pushed these subtypes of MM into relative incomprehensibility and disinterest. Since there is limited knowledge about pre-MM lesions and a scarcity of corresponding molecular pathological biomarkers, early diagnosis, as well as early intervention, are very challenging, leading to the short life expectancy of the affected patients. Further studies, tailored to the oncogenic pathways and driver mutations, are required to improve the overall outcomes of these scary malignancies, especially now that the era of precision and personalized oncology is commencing in the field of rare cancers. Currently, the multidisciplinary approach, the centralization for the cure in centers with high expertise and the inclusion in clinical trials, whenever possible, are warmly recommended.

## Figures and Tables

**Figure 1 biomedicines-10-00150-f001:**
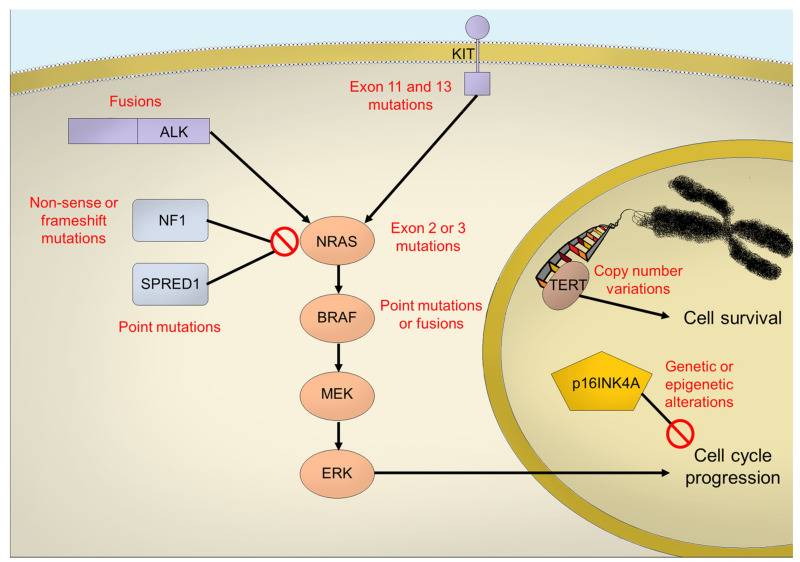
Representation of the most frequent genetic alterations in a mucosal melanoma cell and the principal molecular pathways involved.

**Figure 2 biomedicines-10-00150-f002:**
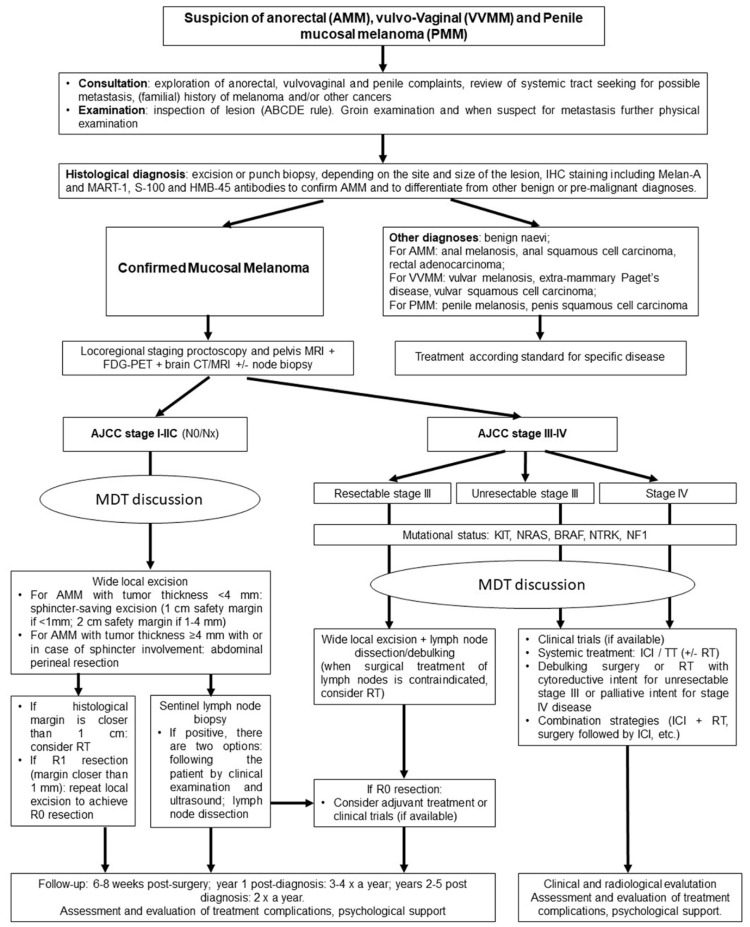
Flowchart management of anorectal, vulvo-vaginal and penile mucosal melanoma. CT, computerized tomography; FDG-PET, fluorodeoxyglucose positron emission tomography; ICI, immune checkpoint inhibitor; MRI, magnetic resonance imaging; RT, radiotherapy; TT, targeted therapy.

## Data Availability

The corresponding author will provide the data or will cooperate fully in obtaining and providing the data on which the manuscript is based for examination by the editors or their assignees.

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
