# Peer review of "Anorectal and Genital Mucosal Melanoma: Diagnostic Challenges, Current Knowledge and Therapeutic Opportunities of Rare Melanomas"

_biomedicines, 2022, doi:10.3390/biomedicines10010150_

Round 1

Reviewer 1 Report

Dear Authors,

This is an interesting paper since mucosal melanoma is a hard to treat cancer, very challenging for practitionners.

My remarks:

Line 229/230: "tumor thickness 1-4 cm " and ">4cm" --> did  you mean mm instead of cm?

Figure 2 : Flow chart is barely legible. Maybe removing the "Other diagnosis" from the chart would give a better overview of the proposed startegy. I do not agree with the classification : stage IIIA (AJCC) does not correspond to N0. In my opinion, ICI (or TT) should systematically be proposed as an adjuvant therapy for stage III (N+). This flow chart is not clear regarding this issue: do you recommend only radiotherapy? combined ICI+RT?

Systemic treatments : there are now more data concerning the use of ICI for mucosal melanoma. Especially the recent review of Jiarui et al. Immune checkpoint inhibitors in advanced or metastatic mucosal melanoma: a systematic review. Jiarui L et al. Therapeutic Advances in Medical Oncology. 2020.

I am therefore surprised with the references that you chose, for example why did you cite a 2018 French study that included 15 patients when another French study was  published the same year reporting  123 anogenital melanomas? --> C. Mignard et Al.  Efficacy of Immunotherapy in Patients with Metastatic Mucosal or Uveal Melanoma. J Oncol. 2018

Syntax: line 499, "instead" --> instead of what?

Author Response

This is an interesting paper since mucosal melanoma is a hard to treat cancer, very challenging for practitioners.

Response: We greatly appreciate the First Reviewer's valuable comments and we thank the Reviewer for having grasped the main aim of our manuscript. We have now revised our manuscript to address and accommodate the reviewer’ s comments and suggestions. We have included a point-by-point.

My remarks:

Line 229/230: "tumor thickness 1-4 cm " and ">4cm" --> did  you mean mm instead of cm?

Response: Thank you for noticing, it was a mistake. We have corrected it.

Figure 2 : Flow chart is barely legible. Maybe removing the "Other diagnosis" from the chart would give a better overview of the proposed startegy. I do not agree with the classification : stage IIIA (AJCC) does not correspond to N0. In my opinion, ICI (or TT) should systematically be proposed as an adjuvant therapy for stage III (N+). This flow chart is not clear regarding this issue: do you recommend only radiotherapy? combined ICI+RT?

Response: Thank you for your comment. We have modified the flowchart and we hope to be legible now. We have kept the “other diagnosis” section because it is important to stress the possibility of other diseases miming the melanoma. We have corrected the classification in stage I-IIC and III-IV to avoid misunderstandings, we have also added the possibility of combining RT and ICI.

Systemic treatments: there are now more data concerning the use of ICI for mucosal melanoma. Especially the recent review of Jiarui et al. Immune checkpoint inhibitors in advanced or metastatic mucosal melanoma: a systematic review. Jiarui L et al. Therapeutic Advances in Medical Oncology. 2020.

Response: We would like to express our great appreciation to the first reviewer for her/him careful and precious observations on our manuscript. We have now revised our manuscript to address and accommodate the valuable reviewer’s suggestion. The manuscript’s bibliography has been updated including the suggested article. Please, see the tracked changes version of the resubmitted manuscript.

I am therefore surprised with the references that you chose, for example why did you cite a 2018 French study that included 15 patients when another French study was  published the same year reporting  123 anogenital melanomas? --> C. Mignard et Al.  Efficacy of Immunotherapy in Patients with Metastatic Mucosal or Uveal Melanoma. J Oncol. 2018

Response: We greatly thank the reviewer for this observation, we added the reference in the paragraph. The two little studies were included in our review since they analysed the different impact of anti-PD-1 and anti-CTLA-4 in VVMM patients, therefore being of interest for us.

Syntax: line 499, "instead" --> instead of what?

Response: Thank you for noticing it, it was a typo, we have removed it from the text

Reviewer 2 Report

Dear colleagues,

I was delighted to review your extensive literature search on such an interesting topic.

I think it could be relevant for clinicians in clinical practice to be aware of different patterns of metastases (for example ocular melanoma --> liver metastasis therefore additional MRI can be useful, sinonasal melanoma --> lung metastases, therefore CT scan of lungs with thin slices can be useful). Therefore I would suggest to be a bit more specific onwards imaging in case of extended disease with the metastatic patterns, when suitable. Furthermore, FDG-PET/CT has been validated for staging, treatment response assessment and post treatment surveillance in case of cutaneous MM in the European Guidelines. Therefore, I would also suggest to review literature on hybrid imaging in the listed melanoma, particularly in context new treatment agents, such as immune-checkpoint-inhibitors.

Finally, I would suggest to correct spelling mistakes through the manuscript.

Looking forward to reviewing the revised manuscript.

Best wishes,

Author Response

Reviewer n.2

Dear colleagues,

I was delighted to review your extensive literature search on such an interesting topic.

Response: We sincerely thank the Reviewer for the time and kind consideration of this manuscript. We have now revised our manuscript to address and accommodate the reviewer’ s comments and suggestions. We have included a point-by-point response.

1) I think it could be relevant for clinicians in clinical practice to be aware of different patterns of metastases (for example ocular melanoma --> liver metastasis therefore additional MRI can be useful, sinonasal melanoma --> lung metastases, therefore CT scan of lungs with thin slices can be useful). Therefore I would suggest to be a bit more specific onwards imaging in case of extended disease with the metastatic patterns, when suitable.

Response: We thank the reviewer for this suggestion. We have added new periods in the text using the following references for AMM (Nam et al) and VVMM (O’Regan et al). Please, see the track-changes of the uploaded manuscript.

2) Furthermore, FDG-PET/CT has been validated for staging, treatment response assessment and post treatment surveillance in case of cutaneous MM in the European Guidelines. Therefore, I would also suggest to review literature on hybrid imaging in the listed melanoma, particularly in context of new treatment agents, such as immune-checkpoint-inhibitors.

Response: We greatly appreciate this careful comment. Unfortunately, to date, there is very few data about the use of FDG-PET/CT in mucosal melanoma. However, we have added two references (Seban et al; Zukotynski et al) to address the potential use of this technique in mucosal melanoma.

3) Finally, I would suggest to correct spelling mistakes through the manuscript.

Response: Done, thank you for noticing. Please, see the track-changes version of our manuscript.
